# Developing and Managing Film-Related Tourism in the All-for-One Model at a Tourism Destination: The Case of Hengdian Town (China)

Xin Cui 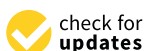

China Tourism Academy (Data Center of the Ministry of Culture and Tourism), Beijing 100005, China; xincui1995@126.com

**Abstract:** All-for-one tourism is a new planning concept proposed by Chinese tourism scholars and practitioners, which has been formally regarded as a new tourism model to develop the country's tourism industries since 2016. It aims to stimulate the growth of the tourism industries across the entire region, encompassing all tourism and tourism-related resources at a destination. Hengdian Town is a tourism destination in China that has implemented the model of all-for-one tourism to develop and manage its film-related tourism. Based on the data collected from ethnographic methods (participant observations and individual interviews) in Hengdian Town, this paper examines the ways that the destination manages its film-related tourism, as well as the outcomes of such approaches, by applying the model of all-for-one tourism. The findings reveal that Hengdian Town has leveraged the model to manage its tourism resources and provided tourists with a comprehensive travel experience. This paper also explores the benefits and drawbacks of managing film-related tourism in the all-for-one tourism model. By focusing on film-related tourism, this study provides a unique perspective on the all-for-one tourism model.

**Keywords:** all-for-one tourism; film-related tourism; tourism policy; film studio; tourism management; ethnography



## 1. Introduction

In 2010, the concept of all-for-one tourism was first introduced from an academic perspective by Chinese scholar Xiaoran Hu, based on a case study of tourism management in the city of Dalin, China. In August 2015, the model of all-for-one tourism was one of the core topics at the National Conference on the Tourism Business (China) discussed by Chinese tourism management practitioners. The discussions revolved around the implementation of this model on a national level in China. In 2016, the Chinese government formally decided to apply the model of all-for-one tourism to develop and manage the country's tourism industries [1]. This model of tourism is characterised by a focus on developing tourism across a broad range of local areas and places, rather than focusing on a single attraction, and aims at developing a tourism project in partnership with all stakeholders and involving the overall planning and cooperative development of all tourism-related industries [1–7]. More tourism resources are utilised to develop the tourism industries, and more individuals are encouraged to participate in the tourism industries. The implementation of the all-for-one tourism model in China contributed to the country's transformation and upgrading of tourism [1,5,8–12]. According to the 'Report on the Development of Cultural Tourism in China 2017', published by the China Tourism Academy, in 2016, the year that China started to apply the model of all-for-one tourism, the economic contribution of the tourism industries, especially cultural tourism, to the national economy has continued to grow and strengthen. Here, cultural tourism refers to people's journeys to certain cultural attractions with the purpose to satisfy their travel needs and interests in cultural elements, activities, and events [13,14]. Film-related tourism is a form of cultural tourism

that involves travelling to destinations and attractions with connections to film-related elements such as film products, arts, culture, businesses, celebrities, events, and more. Here, the term 'film' refers to film (movie), television, and the representations of other screen media [15]. In film-related tourism studies, screen media in all its forms can have the potential to motivate and catalyse people to travel.

Hengdian Town, located in the south-central area of Dongyang City (county-level city) in Zhejiang Province, China, has applied this model to develop and manage its film-related tourism at a town-wide level since 2016 [16]. It is home to Hengdian World Studios, the world's largest outdoor shooting base and film studio theme park, owned by the Hengdian Group. Since 2016, the town has implemented the all-for-one tourism model to develop and manage its film-related tourism in accordance with national policy [16]. By integrating all potential and possible resources, the town has effectively created a tourism brand for the entire town as a film-related destination, which goes beyond the mere promotion of the Studios as a tourist attraction.

This study is an empirical investigation of how Hengdian Town manages and develops its film-related tourism in the all-for-one model at the town-wide level. Ethnographic methods, including participant observation and face-to-face interviews with a local official at the governmental place branding institute, were applied from 2018 to 2021 for collecting data. The research examines the ways and outcomes of applying the model of all-for-one tourism to develop and manage film-related tourism based on the case of Hengdian. It will show the film-related tourism activities and products tourists can participate in and consume, as well as the film-related objects and people they can observe and interact with in Hengdian. Additionally, this study will evaluate the development and management of film-related tourism within the all-for-one tourism model. This research makes important contributions to the literature on film-related tourism by introducing a new development and management model and evaluating the advantages and disadvantages of this model.

This paper will first provide an overview of the key concepts and terms related to film-related tourism and all-for-one tourism. Subsequently, the research methods used in the study and the research setting will be introduced. The paper will then present the discussion and findings of the study, divided into two sections. The Section 1 will analyse the features of all-for-one film-related tourism at Hengdian World Studios, the main film-related tourism attraction in Hengdian Town. The Section 2 will investigate all-for-one film-related tourism in the residential and public areas of the town, showcasing the wide range of tourism products and activities offered. By comprehensively examining the development and management of film-related tourism in Hengdian within the all-for-one tourism model, this study aims to offer insights into how destinations can utilise film-related tourism to provide distinctive and captivating tourism experiences. Finally, based on the case of Hengdian, it will evaluate the benefits and drawbacks of the all-for-one tourism model.

## 2. Conceptual Background

### 2.1. Film-Related Tourism

Over the last three decades, there has been a growing interest in research in the field of film-related tourism [17]. The concept of film-related tourism gained significant attention in the tourism research field by the early 2000s. Researchers largely relied on case studies to obtain knowledge about this topic [18]. Film-related tourism can be seen as a rapidly expanding cultural phenomenon on a global scale, stimulated by the growth of the entertainment industry and the increasing ease of travel worldwide [15].

A number of relevant definitions and terms of film-related tourism have been introduced in previous research, for the purpose of this article, the term 'film-related tourism' will be applied to refer to people's journeys to all film-related attractions and sites. This terminology is adopted because it expands the focus of film-themed attractions at the destination to all-related places that may introduce the culture and history of film and film-related industries, show film-related elements, provide film-related activities and services

or hold film-related events. Meanwhile, when using 'film-related tourism', not only the power of film in motivating people's journeys to film-related sites is highlighted but also the influences of other (film-related) screen media are included. The term 'film-related tourism' is appropriate to be generalised and used in such a situation, where the on-site film-related elements, such as stories, services, activities and facilities, can play certain and specific roles in stimulating film tourists' visits to a destination [19–21]. Film premieres, film festivals, and film museums, for instance, can be seen as the typical film-related elements that are capable of generating tourism at a destination [21]. Thus, the definition of film-related tourism in this paper is stated as a cultural phenomenon in that people visit a site as a result of it having been featured in a moving image and/or involving and representing film-related features, stories, culture, and products on-site.

*2.2. All-for-One Tourism*

All-for-one tourism essentially is a new planning concept proposed by Chinese tourism management practitioners. It is also regarded as a new tourism model to develop the country's tourism industries [1,10,12]. The ultimate aim of all-for-one tourism is to attain a scientific and reasonable integration of resources, industries, and developments, along with social co-construction and sharing by optimising and upgrading the complex systematic structure of a specific tourism region [5,7,8]. The concept of 'all-for-one' is introduced and applied here for highlighting the differences between the new tourism model and the scenic spot tourism model in the process of developing and managing the tourism industries in China. When implementing the scenic spot tourism model, which involves promoting and developing specific tourism destinations, the construction and management of closed scenic spots are carried out in a fragmented manner, leading to the independent development and management of these specific tourism sites [1]. A tourist destination place is plotted out into several single separate and isolating tourism sites. In addition, scenic spot tourism has sunk into difficulty in driving the development of the regional economy over the past decades in China [5,11,22]. The all-for-one tourism model has been put forward as a new approach and mode to managing and developing tourism, which treats a tourism area as a complete destination, enabling coordinated planning, integrated marketing, and holistic management to enhance the tourism destination's overall appeal [1–5,8,9]. Thus, the all-for-one tourism model can be seen as a higher-level tourism mode compared with the scenic tourism model [23]. In 2018, the General Office of the State Council (China) published a national document to provide strategic guidelines for all regions in the county to develop and manage the tourism industries in the way of the all-for-one model—'Guiding Opinions of the General Office of the State Council on Promoting the Development of All-for-one Tourism'. It refines the specific and concrete operations in developing all-for-one tourism or applying the mode of all-for-one tourism in China. This includes promoting cooperation among the tourism industries and other relevant industries, shifting from closed tourism circles to open collaboration, and moving from developing single tourist attractions to providing comprehensive destination services. Tourism destinations should make efforts to stimulate the effective integration of various relevant industries, the concerted cooperation of various relevant departments, the participation of the whole destination residents, and the full use of all the destination attractions and sites [2]. It follows that tourism destinations should integrate 'all' available landscapes, 'all' workable time, 'all' relevant industries and 'all' potential people [2,5] to establish a tourism culture where people adopt a tourist mindset and create a tourist-friendly environment in all locations [1,6,9,10]. In the case of Hengdian Town, no longer applying the model of scenic spot tourism, the all-for-one tourism model has been applied and implemented in the constant development of new tourism areas and the integration of different tourism and tourism-related industries since 2016.

## 3. Research Setting and Methods

### 3.1. Research Setting

Hengdian Town, under the jurisdiction of Dongyang City (county-level city), is located in the south-central area of Jinhua City (superior province-level city), Zhejiang Province, in China (Figure 1). It is the location town of Hengdian World Studios, the world's largest outdoor film studio and film shooting base, which occupies 6.67 square kilometres in the town. Launched in 1996, Hengdian World Studios had around 130 indoor film studios and more than 10 outdoor filming areas and film-themed tourism attractions by 2020 [24]. These outdoor film studios reconstruct the landscapes, streetscapes, and imperial and folk buildings in a number of Chinese past dynasties, as well as in some early-modern Chinese cities, spanning centuries of Chinese history, and they simulate a number of real (existing or vanished) heritage sites in different areas of China. The theme of each outdoor film studio is designed based on the architectural style and cultural characteristics of a certain Chinese past dynasty from 221 B.C. to 1912 C.E. or a certain Chinese early-modern city in the 1910s–1940s. They can satisfy different media productions and crews to shoot their screen media works with different historical backgrounds and storylines. Tourists can also experience film-related tourism at different tourism sites involving different film and cultural heritage tourism features by participating in and consuming various types of tourism activities and products. In 2010, the Studios was also classified as the highest-level tourism attraction in China by the China National Tourism Administration (now the Ministry of Culture and Tourism). Since 2016, the town of Hengdian has started to apply the model of all-for-one tourism to develop and manage its tourism industries, and Hengdian World Studios is the core tourism attraction at the destination [16]. In 2020, the town generated more than CNY 20 billion (approximately GBP 2 billion) in tourism revenue while also accommodating over 20 million tourists [25].

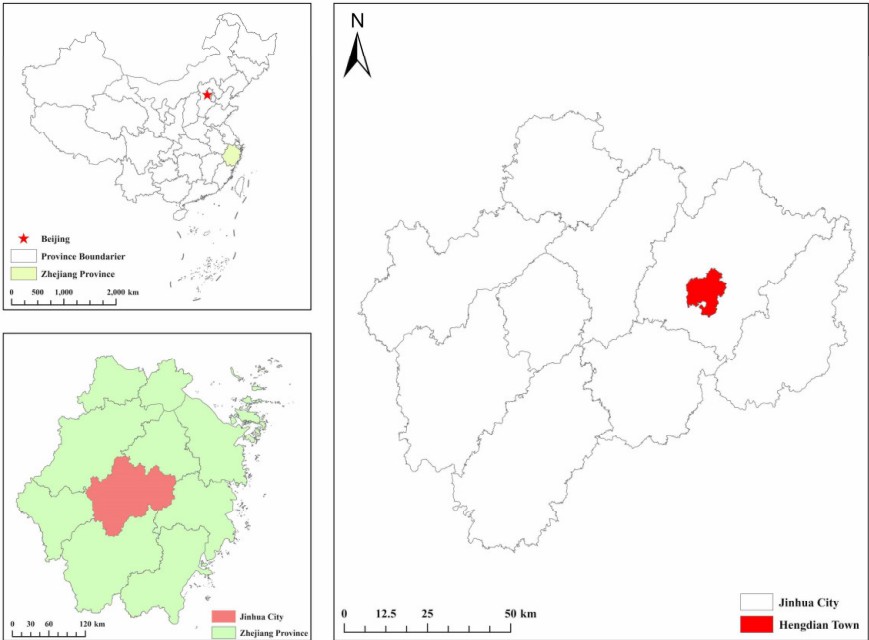

**Figure 1.** Localisation of Hengdian Town.

### 3.2. Ethnographic Methods

The methods that were applied for collecting relevant data and information are ethnographic methods. Data presented in this paper were collected and consolidated from ethnographic visits undertaken in the town of Hengdian in both high and low tourism seasons from 2018 to 2021. The ethnographic methods conducted in this study include my participant observation in the journeys to Hengdian Town as a tourist and formal interviews with an official of the local governmental place branding institute. It follows that

some of the substantive data presented in this paper were collected and consolidated from my participant observation in Hengdian Town. Ethnographic interview data presented in this paper were collected from the face-to-face interviews with Gang Zhang, the director of the Management Council of Hengdian Film and Television Cultural Industry Experimental Zone, in Hengdian Town in August 2019 and October 2020. The zone is a specific administrative region to develop and manage the film and television industries in Hengdian, where all film and television production companies and film-related businesses are administered uniformly in a centralised way by the local government. The interview contents will show why and how Hengdian develops and manages its film-related tourism in the all-for-one tourism model. In the process of data collection, ethnography involves researchers' overt or covert participation in (other) people's daily lives for a period to observe and interact with people in the field, and thus researchers are able to draw on a range of sources of data through participant observation [26]. It follows that in the case of Hengdian, data and knowledge about the tourism destination, tourists' on-site experiences, and the local people have been collected and gained through my participation in local touristic activities and events, my interactions with other people on site, and my self-observations and observations of on-site people and the destination. Meanwhile, ethnography mainly involves (but not necessarily) spending a lengthy period of time living among the local people and having a social relationship with the local communities under study [27]. Based on this, by doing ethnographic methods in Hengdian, I was able to observe and interact with local residents and tourism stakeholders to know their attitudes and perceptions about the local tourism development and the impacts of film-related tourism on Hengdian. That is to say, the method of ethnography has allowed me to understand the destination and its tourism industries from the perspectives of local people and tourists. Their narrations about how film-related tourism impacts their daily lives and work in the past and at present particularly have contributed to my understanding of Hengdian's all-for-one film-related tourism. In the case of Hengdian, the value and significance of doing ethnography in the data collection process are threefold: (a) understanding Hengdian's film-related tourism contents (products, activities, events, and so on); (b) exploring the ways and outcomes that Hengdian develops and manages its film-related tourism by applying the all-for-one tourism model; and (c) examining the benefits and drawbacks of the all-for-one tourism model based on the participant observations and ethnographic interview contents in Hengdian.

## 4. Results and Discussion

The discussion of the ethnographic experiences will be separated into two sections based on a chronological order of the visits that took place from 2018 to 2021 to Hengdian Town. The Section 1 will show our travel experiences and findings during the first and second ethnographic field visits in 2018 and 2019, as the main research and data-collecting areas in these two field trips are inside Hengdian World Studios. The Section 2 will show my travel experiences during the third and fourth ethnographic field visits in 2020 and 2021, and the main researching and data-collecting areas in these two field visits are in the public and residential areas outside the Studios.

### 4.1. All-for-One Film-Related Tourism in Hengdian World Studios

The implementation and application of the all-for-one tourism model in Hengdian World Studios are reflected in the use of almost 'all' film-related resources, businesses, objects, elements, and people to design and organise film-related tourism products and activities. These include (a) representing and showcasing film-related decorating elements and objects; (b) providing film industrial tourism products and activities; and (c) offering various film-related tourism products and activities. This paper will now explain the implications of my experiences and observations.

Firstly, as the film locations of more than 3200 screen media products [28], the film settings, backdrops, and props as the decorations in the studios are significant film-related tourism products. During the ethnographic visits to the studios, I observed the staged

photos, posters, and other visual information in some film locations, which to some degree worked as tourism signposts to direct and remind tourists of the connections between these film locations and certain film and television works made there. Taking 'The Palace of Emperor Qin' film studio as an example, it is the film location of several Chinese domestic film and television works, such as Emperor and the Assassin (Kaige Chen, 1998), Hero (Yimou Zhang, 2002), Pained Skin (Gordon Chen, 2008), and The Untamed (Weiwen Zheng and Jialin Chen, 2019). During the journeys to this film studio, I observed the staged photos, posters, props, and staged film settings around the film locations of some films and television dramas (Figure 2). In certain film locations, tourists can find billboards featuring staged photos of the main characters from particular films and television dramas that were filmed and made in the Studios. These billboards also provide information about where the scenes were shot. Some tourists took photos of the billboards as a memento of their film journeys. This conforms to Sue Beeton's viewpoint that film tourists gather memorabilia related to locations, actors, and characters, bringing them back home along with anecdotes of stardom that elevate their status among their acquaintances [21]. These film settings and locations are not only provided to media crews for their filmmaking works but also used as tourism attractions for tourists to visit. The all-for-one tourism model is applied by the Studios to manage film-related tourism by utilising 'all' available spaces within their tourist attractions and 'all' available resources to showcase its film culture and stories and offer film-related tourism products. A series of film-related intangible elements (e.g., culture and stories) were transferred into tangible film-related tourism products and objects that tourists can see and interact with.

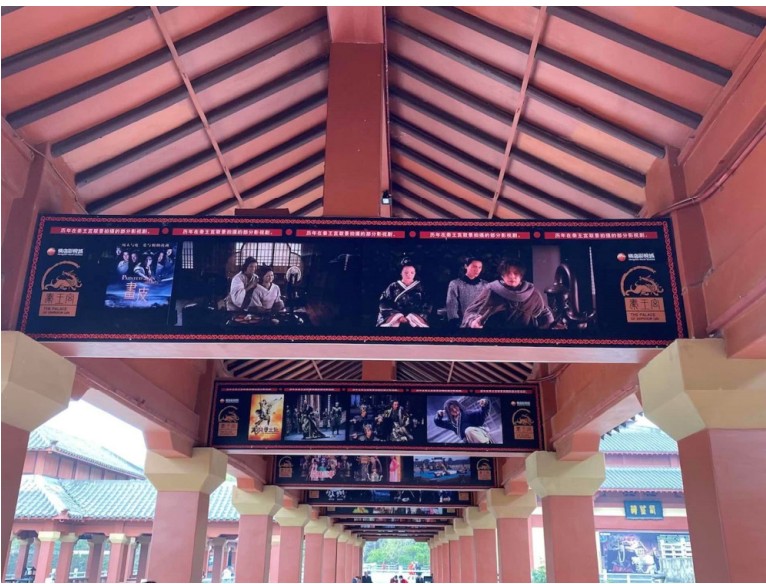

**Figure 2.** The Chinese words shown in this image are the background information of these staged photos and posters. Translation: Staged photos and posters of screen media products made in 'The Palace of Emperor Qin' film studio, Hengdian World Studios (photo by the author, 2020).

Secondly, as a working film studio and film theme park, the Studios also presents the actual ongoing works of film and television productions and media crews. In this regard, developing and managing film-related tourism in the all-for-one tourism model refers to the commercialisation of 'all' potential people in the film and television industries (film celebrities and media crews) as film-related tourism products. Tourists at Hengdian World Studios can experience the authentic film production process on-site. Such an advantage to a large degree results from the Studios' operation mode, as media crews and on-site tourists share the same space. During my first journey to the Studios in 2018, I was fortunate to get a chance to observe the ongoing work of a media crew in the 'Qing Ming Shang He Tu' film studio. Although tourists in the Studios have relatively high freedom to go

sightseeing, wander around the film locations, and observe the crews and film celebrities working at the filming locations, the travel routine is still limited by the filming activities through fencing off the filming areas with security to prevent tourists' access. One of the core tourist activities in (industrial) film studio theme parks is to visit the backstage areas of filmmaking [21]. Observing the ongoing works of media productions helped satisfy tourists' curiosity about what it was like to film a scene. The media productions at work to some degree can function as a kind of industrial tourism product, allowing tourists to observe the production process of screen media work and serendipitously or intentionally encounter, observe, and take photos of them in their travel routines. It follows that tourists can encounter film celebrities at work, or off work, in Hengdian World Studios. The film celebrities and crews at work are not originally designed as tourism products, as the actual reason they place themselves in the studios is for filmmaking rather than for developing the tourism industries. However, the all-for-one tourism model aims to use 'all' possible people to develop and manage its tourism industries and to make 'everyone' become the tourism image [1,6,9,10]. The film-related people (crews and celebrities) in the film and television industries are also used as important tourism resources, and observing on-site media productions' filming activities is designed as one of the important film-related tourism activities in the studios.

Thirdly, based on my ethnographic observations, Hengdian World Studios also uses 'all' possible settings, facilities and people to provide tourists with more immersive travel activities and tourism-related services, which it has been doing since 2018. During my third visit to Hengdian in October 2020, in which the main purposes of the visit were to conduct a face-to-face interview with the local place branding institute and to observe the representations of film elements in the town of Hengdian, I was introduced by the place branding institute to a new 'immersive experience' of film-related tourism and related touristic activities and services as well as a new 'night view' and 'neon light show' in the 'Guangzhou Street · Hong Kong Street' film studio.

Following the recommendation of the local governmental institute in the interview in October 2020, I re-visited the 'Guangzhou Street · Hong Kong Street' film studio for the 'immersive film-related touristic experience'. The first time I visited this film studio was in April 2018, whereas the 'immersive' film-related touristic activities and services were provided to tourists in October 2018. These were new film-related tourism elements resulting from the 'upgrade' of the tourism site. After the upgrade, the closing time of the film studio was extended from 17:00 to 20:00 in order to provide tourists enough time to join the night tour at the 'Hong Kong Street' touristic area to see the light show involving neon lamps on the film settings and buildings and night outdoor live performances. The touristic atmosphere at night created by the light show and performances in this film studio is indeed different from that at daytime, as it shows Hong Kong's night life in the 1910s. During the night tour, I also discovered the ways that the film studio had upgraded in order to provide tourists different film-related touristic experiences. Firstly, the film studio re-constructed its film settings and backdrops as well as the locations of some famous film and television works made in the 'Hong Kong Street' area, such as the 'Hua Dong Photo Studio' in the television drama The Disguiser (2015), which is now reproduced in the 'Guangzhou Street · Hong Kong Street' film studio and designed as a souvenir store to sell paintings and photos with themes of traditional Hong Kong in the 1910s. Through visiting the film settings and locations, I was brought into a fictional world in which the settings and buildings were designed after fictional scenes from film and television works. Secondly, in order to emphasise the 'reality' of the fictional environment, the studio also launched a film-themed hotel, in which the rooms were designed after fifteen film and television works made at the Studios, and where the hotel guests can dress in character costumes to interact with each other. 'Become a film and television character and stay in a film or television work' is the slogan of this hotel. In this way, tourists are encouraged to play the roles of film and television characters when living in the hotel.

In addition, during my third ethnographic journey to Hengdian, I also visited a new tourism attraction (film studio)—the 'Legend of Bund' film studio, which is an outdoor film studio designed to replicate the Chinese city of Shanghai in the 1930s and 1940s. Different from some other film studios, such as the 'Guangzhou Street · Hong Kong Street' film studio, which was originally built for making a specific film, the 'Legend o Bund' film studio, which opened in late 2019, was built for both film-making and tourism purposes. Thus, the site provides a range of tourism-friendly facilities and services, such as tourist sightseeing trams, a film character dress rental service, and a make-up service.

Finally, tourists in the Studios were also able to visit film-related tourism attractions or participate in tourism activities that are not specifically relevant to any screen media works. One of the core characteristics of developing and managing tourism industries in the all-for-one tourism model is to fully use the tourism attractions and sites [2]. In the case of Hengdian, Hengdian World Studios serves as the core film-related tourism attraction in the town, utilising 'all' available indoor space and areas to construct and introduce diverse facilities and organise various activities and events related to film-related tourism. This allows tourists to partake in various types of film-related tourism in a multitude of ways.

Tourists were able to watch several live stage performance shows in different film studios and tourism attractions, which are performed for free to tourists daily at regular times and combine Chinese traditional culture and historical stories with various film techniques. It follows that the way the Studios applies the all-for-one tourism model is to use artificial and constructed filming settings as important physical tourism locations to develop and manage its film-related tourism and offer multiple types of film-related tourism products. During my visit to the 'Palace of Ming and Qing Dynasties' film studio in 2018, I had the opportunity to watch the live stage show 'Secret Story Happened in Qing Court'. This 20-min live performance showcases the historical tales of Qing Dynasty emperors in the Forbidden City (Beijing, China), with a theme and style that aligns with the studio's focus. In this guise, film settings and facilities are used variously as tourism locations for the Studios to provide different film-related tourism products and services and to organise film-related tourism activities and events.

### 4.2. All-for-One Film-Related Tourism in the Residential and Public Areas in Hengdian

The data represented in this section were collected from the ethnographic visits to the residential and public areas in the town of Hengdian in 2020 and 2021. The ways that the town implements the all-for-one tourism model include (a) the integration of film elements into the town's basic facilities and the representations of film elements in the public areas and (b) the encouragement of residents' participation in film-related tourism businesses. This paper will now explain the implications of my experiences and observations.

One of the significant observations of the destination regarding the town's all-for-one film-related tourism is the integration of film features into the town's basic facilities and the representations of film features in the public areas in the town. Based on my ethnographic observation of Hengdian Town, I observed that various forms of film-related features were presented to tourists, thereby enhancing the content of their on-site film-related tourism experiences. For instance, the town's road and street decorations comprise a collection of film-themed iron and stone sculptures and artworks located throughout both tourist and residential areas, which tourists can observe while wandering around (Figure 3). Tourists can see the film-related decorations whether they are taking a sightseeing bus, joining a walking tour, or exploring the town on their own. Some of these decorations serve practical purposes, such as road signposts and landmarks, while others are purely ornamental, highlighting the town's film-related culture, history, and works. As an illustration, along the 'Film and Television Road' in the town, banners on lampposts display staged photos and posters of popular domestic film and television productions in China that were filmed and produced in Hengdian Town (Figure 4). The roads and streets in Hengdian play a crucial role in promoting the destination's film-related culture, stories and history, contributing to an immersive atmosphere that envelops tourists in the stories and culture of the local film

and television industries. Furthermore, these locations offer significant physical tourism resources that can be utilised to develop and manage all-for-one film-related tourism initiatives, such as creating film-themed walking tours, offering photo opportunities with film-inspired sculptures and artwork, or staging events and festivals that celebrate the town's film history and culture.

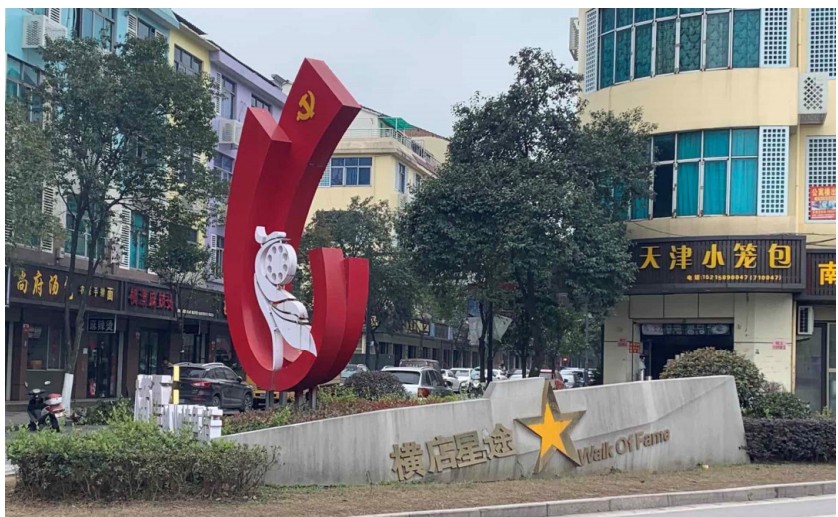

**Figure 3.** Stone sculptures and artworks with film elements (photo by the author, 2020). The Chinese contents shown in this image are the names of restaurants. 'Walk of Fame' is the English translation of the Chinese words represented in the stone.

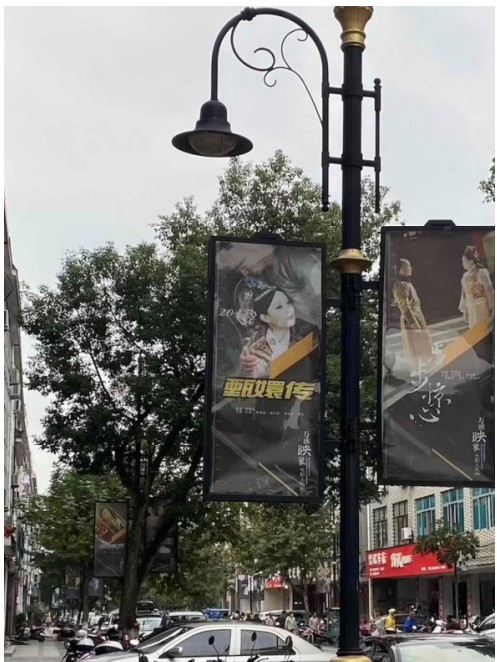

**Figure 4.** Staged photos of screen media products showcased in the lamp post banners in Hengdian Town (photo by the author, 2020). The Chinese contents shown in the lamp post banners are the names of the screen media works. The English names of these two screen media works are 'Empresses in the Palace' (Left-side staged photo showcased in the lamp post banner) and 'Scarlet Heart' (Right-side staged photo showcased in the lamp post banner).

During the ethnographic visit to Hengdian Town in 2020, a face-to-face interview was conducted with the director of a local governmental place branding institute—Gang Zhang.

In the interview with Zhang, when talking about why representing film features in the town's basic facilities in Hengdian, he introduced:

> With the success of Hengdian World Studios in both the film and television industries and tourism industries, the town's government branding and marketing departments have been aware of the popularity of film-related tourism and the significance of effectively developing and managing film-related tourism. We hope tourists in Hengdian experience film-related tourism not only in the Studios but in the whole town. We believe that integrating film elements into the facilities can highlight the characteristics of film-related tourism in Hengdian and improve the place image as a film tourism destination.
>
> (Zhang, 2020)

Zhang's comments imply that one of the tourism development and management strategies and place-branding strategies Hengdian employs is to integrate film-related culture and stories into its place images to stress the specificity of the town's film-related culture and the characteristics of tourism products and services. Through leveraging the popularity and influence of Hengdian World Studios in the tourism industries, other places and areas within Hengdian can be developed into film-themed tourist sites using the all-for-one tourism model. This involves incorporating film-related elements into existing tourist attractions, such as historical landmarks and cultural sites, or creating new film-inspired attractions that cater to the diverse interests of film tourists. Over time, these efforts can contribute to the expansion of Hengdian's film-related tourism offerings, increasing the town's appeal to a wider range of tourists.

In addition to the 'Film and Television Road', a number of other streets and roads in the town were also named with film-related information or have certain relevance to the Studios, for instance, 'Hua Xia Road', the road in front of 'Hua Xia Culture Park' film studio; 'Qing Ming Shang He Tu Road', the road in front of 'Qing Ming Shang He Tu' film studio; and 'Ming Qing Palace Street', the street in front of 'Palace of Ming and Qing Dynasties' film studio. When talking about the ways to integrate film elements effectively and harmoniously into the town's basic facilities, Zhang explained:

> In recent years, the government of Hengdian has put forth a branding policy to develop the town into a cohesive and integral tourism site. As part of this effort, we collaborated with Hengdian World Studios to design and construct public areas throughout the town that share similar themes and styles to those found in the Studios. To further reinforce the film studio's thematic influence on its surrounding public areas, basic facilities such as the sightseeing bus station in the vicinity of 'The Palace of Emperor Qin' film studio have been constructed and designed in a Qin-style architecture, adding to the immersive experience for tourists.
>
> (Zhang, 2020)

Conforming to Zhang's comments, during the ethnographic visits to Hengdian Town in 2020 and 2021, I also observed the film-related silhouettes and paintings on the street walls and pavements, such as filmmaking props and equipment, filmmaking activities, and film-related characters (Figure 5). Moreover, film elements are also integrated into the town's basic facilities, for example, the shape of bus station boards is designed as a film clapper board and the shape of public billboards is designed as a film roll. When I stood at a bus station to wait for the bus to our hotel, a Chinese female tourist surprisingly remarked to her fellows, 'Look! It is interesting to see the station board in the shape of a clapper board!'. Her fellows also responded, 'That is why Hengdian is the "home of film"'. As the all-for-one tourism model aims to make everywhere the tourist environment and to integrate 'all' available landscapes to develop the tourism industries [1,6,9,10], in the case of Hengdian, more residential and public areas are used as tourism areas or integrated with film-related features for developing the town's film-related tourism and offering tourists more film-related tourism products.

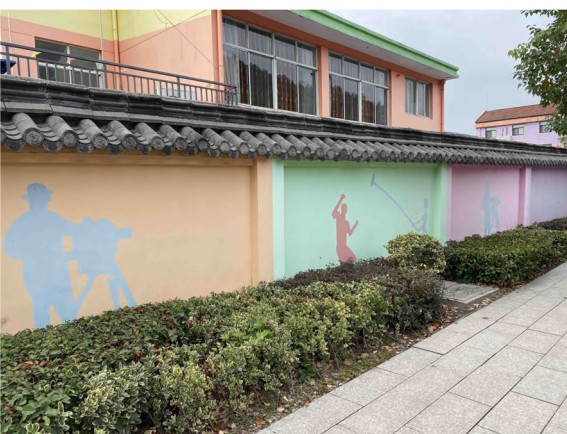

**Figure 5.** Film-themed silhouettes painted on a street wall in Hengdian Town (photo by the author, 2020).

Residents' participation in the film-related tourism businesses also contributes towards branding the town as a destination of all-for-one film-related tourism. The action of the all-for-one tourism model in this guise contributes to making 'everyone' (local people) become a tourist image and 'everywhere' (resident and public areas) become a tourist environment [1]. More available landscapes, more workable time, more relevant industries, and more potential people are involved in the process of tourism development and management in Hengdian Town. Zhang in the interview also provided examples to explain how local citizens are involved in tourism-related businesses:

> Local citizens in Hengdian who own self-built houses have the option to rent out their properties to media crews and tourists independently. To incentivise and recognize their contribution to the tourism industry, the local government offers an allowance of 500 Yuan [approximately 50 Pounds] per bed, per year. Additionally, locals can collaborate with Hengdian World Studios to transform their houses into film-themed hotels for tourism purposes. Some locals even lease their homes to out-of-town businessmen for use as restaurants or hotels. In either case, renting out their properties provides a stable source of income for these individuals.

> (Zhang, 2019)

In terms of the hotel industry, a number of residents re-built and re-designed their houses as privately-owned hotels, especially in the areas around the Studios' tourism sites and attractions. This is possible because hotels located in tourist areas are more profitable than others [29]. Compared with the high-end business hotels and tourist resorts, based on my observation at the destination, residents' privately-owned and self-built hotels and rooms are usually designed and built as three- or four-story buildings with plain facades and decorations and simple plaques and boards. During my journey to Hengdian in 2020, a taxi driver explained 'one of the advantages of this kind of hotel is the cheaper price, and the target guests are not only tourists but also extra actors/actresses, who need to work in different media productions in the long term'. A local resident, Mrs Jin (participant has agreed to use her surname in this paper), who has a self-built four-story house and designed the house as a self-built hotel, in a face-to-face interview with me in Hengdian, described the benefits she had received by participating in the tourism business:

> The town government of Hengdian actually encourages us to re-build our houses for tourist accommodation in our lands through, for example, allowing us to apply for a large loan from banks for building the houses and also providing us a low loan rate. In tourist seasons, a large number of tourists chose my self-built house for their accommodation. Even though in off-seasons, some out-of-towners who work at HWS as extra actors/actresses also rent my rooms long term.

(Jin, 2021)

In addition to the hotel industry, individuals can economically benefit from the restaurant industry as well. Mr Liu (the participant has agreed to use his surname in this paper) was a local restaurant owner who had operated his restaurant for 10 years in Hengdian, in a place that was geographically close to 'The Palace of Emperor Qin' film studio, and someone I interviewed at his restaurant after having lunch during my fourth ethnographic field trip to Hengdian in 2021. Regarding the impacts of film-related tourism on the destination from his perspective, in our face-to-face interview, Liu stated:

> Some of the local people, especially those whose houses are geographically near to HWS, including myself, re-designed and re-decorated their houses as restaurants featuring local delicacies and cuisines. In tourism seasons, such as national holidays, my restaurant was always full of tourists and visitors, and thus I could receive high economic returns. As I know, some local residents also prefer to rent their houses as restaurants, and the house rental is one of the resources of their income. However, a group of local residents who have sold their agricultural lands to the Studios but cannot benefit from tourism-related industries, had to move to other cities.

(Liu, 2021)

Even though a number of local people do not directly participate in the tourism industries, the popularity of film-related tourism and other forms of tourism promote the development and growth of the local hotel and restaurant industries and provide individuals more opportunities to become stakeholders in tourism-related industries and profit from their participation. For these individuals, especially local residents, their land and houses can be seen as sources of economic benefits and profits from tourism-related businesses and activities. Mr Shen (the participant has agreed to use his surname in this thesis), a manager of a local hotel (Yilai Boutique Hotel) where I stayed in 2019, 2020, and 2021 when conducting ethnographic research in Hengdian, explained the following in our face-to-face interview with me in 2021:

> Almost all my colleagues and I are the local citizens and residents. In recent years, a number of local young people have decided to stay in Hengdian or the neighboring areas and worked in tourism-related industries. This could be because impacted by the success of the film and television industries and the tourism industries, Hengdian now provides many employment opportunities with great benefits and salaries. A number of young people from the neighboring areas also come to Hengdian and work as Uber drivers and tour guides. Certainly, a number of people, who do not want to work in the tourism and tourism-related industries, moved to other places to work and live.

(Shen, 2021)

The participation and involvement of local residents in tourism businesses and activities can promote and stimulate the development of the tourism industries and relieve the pressure of limited tourism resources in the town, such as land, labour, and space. Using accommodation and restaurants as examples, as of the end of 2017, Hengdian Town has boasted a total of 317 hotels offering approximately 16,120 beds, as well as 298 bed and breakfast accommodations and homestays provided by residents with around 4628 bed spaces [14]. It means about one-fifth of accommodations (bed spaces) in the town are provided by local individuals, which can increase the town's capacity for tourism and give more choices to tourists. This also conforms to one of the characteristics of all-for-one tourism—the integration of 'all' relevant industries—to develop and manage the tourism industries. In the case of Hengdian, the hotel industry is one of the relevant industries that is integrated with the tourism industries for stimulating the rapid development of local film-related tourism. By applying the model of all-for-one tourism, the town's tourism industries are more conducive and tourism-friendly to local people. This model primarily

benefits a specific group of residents whose self-constructed homes are either conveniently situated near the town's major tourist attractions or are easily accessible to tourists.

*4.3. Benefits and Drawbacks of the All-for-One Tourism Model*

Based on the case of Hengdian Town, this paper suggests several advantages and drawbacks of developing all-for-one film-related tourism at a tourism destination.

Firstly, from a tourist's perspective, all-for-one film-related tourism presents a unique way to explore a destination. Tourists' film-related travel activities and experiences are no longer limited to specific tourist attractions, but their travel routes are extended to the whole destination. This means that a destination's film-related tourism can offer more than just a few specific film-related scenic spots; it can encompass entire areas of the destination. The selling points of the destination's film-related tourism can start from several specific film-related scenic spots and extend to other areas of interest within the destination. Hengdian's all-for-one film-related tourism exemplifies this trend, as it proves that film journeys need not be limited to the core tourism attraction—Hengdian World Studios. Instead, tourists can undertake these journeys continuously outside of the Studios. All-for-one film-related tourism provides a unique way for tourists to explore and experience a destination. By extending the scope of film-related activities beyond specific attractions, tourists can gain a more comprehensive understanding of the destination's film culture and history.

Secondly, the Hengdian case suggests that from the perspective of tourism destinations, adopting the model of all-for-one tourism brings more benefits to more residents. By allowing tourists to explore various tourist attractions associated with film-related tourism rather than restricting them to a single area, the all-for-one tourism model could alleviate several issues caused by tourism, such as overcrowding and traffic congestion at certain attractions. The growth of all-for-one tourism results in the utilisation of additional regions within the destination as film-related tourism sites. This expansion enables tourists to explore a wider range of geographical areas beyond the Studios, thereby diversifying their travel routes and movements. This paper indicates that developing all-for-one film-related tourism allows more residents to be beneficiaries of tourists' activities and consumption of tourism products and tourism-related services in the town. More residents can become stakeholders in local film-related tourism by participating in tourism businesses. Engaging with the local population and community in the management and planning of tourism is a fundamental principle of sustainable tourism development, because the success of tourism in a particular region depends on the support of its inhabitants [30]. It is no doubt that all-for-one tourism expands the involvement of local people in tourism industries. In other words, the Hengdian case shows that local people are contributors and beneficiaries of all-for-one tourism.

The third advantage of the all-for-one tourism model is its potential applicability to various film-related tourist destinations across different countries and regions. A destination can develop its all-for-one film-related tourism through, for example, designing more film-related elements, activities, and tourism products at the paths of tourists' movements from one tourist site to the other. The possible film-related tourism destinations that can apply the model of all-for-one tourism are destinations that have film studios or film-themed parks. Hengdian World Studios and some other film studios, such as Universal Studios, Fox Studios, and Paramount Studios, can be classified as the '(industrial) film studio theme park' with the forms and functions of both a working film studio and film-based theme park [14]. Similar to Hengdian Town, destinations that have a film-themed park, such as Disneyland Park, can also develop and manage their film-related tourism through either fully or partly applying the all-for-one film-related tourism. Even though they are launched in different cities and countries, such as Universal Studios in Los Angeles (United States) and Beijing (China), as well as Disneyland Park in California (United States), Paris, (France), and Shanghai (China), considering the characteristics of the all-for-one tourism model, the destinations are also available to apply this model to expand the influence and popularity of these film studio theme park. Taking the Disneyland Park as an example, the theme park

itself can be seen as a typical case of 'fantasy city', John Hannigan's concept [31], which describes a space that has various, sophisticated, and dynamic entertaining themes and special operation formats and forms. There is a clear distinction between the 'fantasy city' and other irrelevant elements; for example, buildings, services, facilities, and activities are excluded. Disneyland Park, as a 'fantasy city', is the theme park that presents its happy regulated vision of pleasure, and it does so appealingly by stripping troubled urbanity of its sting, or the presence of the poor, of crime, of dirt, of work [21,32]. The destinations that have Disneyland Park can apply the all-for-one tourism model to expand the themes of film and fairy tale and other fantastic themes from the Disneyland Park to the other areas of the cities so as to promote the whole destination as a film-related fantasy city. In addition, applying the all-for-one tourism model also encourages the destinations to connect almost all film-related tourist sites, such as film museums, film setting places, and film location places, and provide more film-related elements to tourists at different areas of the city. By doing this, as discussed previously, the cities could extend tourists' travel routes to the whole destination, and more local residents can be benefit from the development of local film-related tourism. It is not essential for a location to fully adopt and advance all-for-one tourism in all its regions, with the integration and cooperation of numerous industries, departments, and individuals. The all-for-one tourism model presents a fresh outlook for destinations seeking to develop and manage their film-related tourism. The new perspective suggests that one of the ways for a destination to provide more film-related tourism products is to maintain the continuity of tourists' on-site film-related experiences rather than to separate tourists' experiences as several travel fragments at the core tourism sites.

Nevertheless, the discussion of Hengdian also implies the potential drawbacks of developing all-for-one film-related tourism. One of the possible drawbacks is the (over) commercialisation of the destination, i.e., the whole destination tends to be a huge tourism product, if considering that what tourists see and experience in a touristic space is staged [33]. Therefore, there is a certain level of risk involved if a tourism destination places excessive emphasis on film-related and tourism-related elements while developing and managing its tourism industries.

Furthermore, developing all-for-one film-related tourism could lead to spatial and labour displacements in Hengdian. As all-for-one tourism aims to create a tourist culture where individuals approach their surroundings with a tourist perspective and a tourist-friendly environment in all locations [1], it inevitably leads to the involvement of more people in the tourism industries and the use of more spaces as tourism attraction areas. Labour thus gradually tends to migrate from other industries to the tourism and tourism-related industries for supporting and benefiting from the development of all-for-one film-related tourism. Some local people may have to leave their home areas in order to make way for local tourism development and look for jobs in other places. More public areas are used in tourism and tourism-related industries, while fewer areas are used in other industries and activities, to some degree providing less development space and opportunity to other industries. Once the destination's tourism industries are impacted by uncontrollable and irresistible external factors, such as extreme climates and pandemic disasters, a relatively large number of workers may need to face the risks of income reduction and unemployment. Therefore, this paper suggests applying different perspectives to see the development of all-for-one film-related tourism. Some film-related tourism destinations can receive benefits through applying this tourism mode and have the ability to withstand the risks and challenges that come with it, while other tourism destinations may not.

The case of Hengdian can reflect why the Chinese government has decided to develop all-for-one tourism at the nationwide level, even though it could bring negative influences to a tourism destination, such as over-commercialisation and social-spatial displacement. The first reason is that all-for-one tourism contributes to the development of a tourism destination's economy and the increase in employment. One of the basic principles of developing all-for-one tourism in China is to maximise tourism benefits [34]. More local areas became tourism sites and environments, where tourists could visit and consume

tourism products, and more local people became stakeholders and practitioners in the tourism industries, thereby benefiting from tourists' consumption. All-for-one tourism also stimulates the development of other industries, such as the hotel industry, restaurant industry, and transportation industry, through, for example, increasing employment in these industries. Secondly, applying the model of all-for-one tourism to develop the tourism industries at a tourism destination could contribute to better representing the destination's tourism culture. In terms of Hengdian, through developing all-for-one film-related tourism, the town had more ways to represent its film culture, for example, using film-themed iron and stone sculptures and artworks as road signposts and landmarks and designing bus station boards as film clapper boards. As a result, the Chinese government decided to promote the development of all-for-one tourism because they have realised that tourism is an effective means to boost the national economy, and all-for-one tourism is an important foothold to boost economic and social development in China.

## 5. Conclusions

This paper has shown how a tourism destination applies the all-for-one tourism mode to develop and manage its film-related tourism based on the case of a Chinese town—Hengdian, in which Hengdian World Studios, the world's largest outdoor film studio and film shooting base, is located. Data were collected from the ethnographic methods during the visits to Hengdian Town from 2018 to 2021. As the core film-related tourism attraction and site in Hengdian Town, Hengdian World Studios employs the all-for-one tourism model to incorporate diverse film-related elements, resources, objects, and individuals into designing and organising its film-related tourism products and activities, thereby successfully developing and managing its film-related tourism. Tourists thus can experience all-for-one film-related tourism by participating in and consuming different tourism activities and products inside the studios. Moreover, the model of all-for-one tourism is also applied in the residential and public areas in the town of Hengdian. Tourists can experience film-related tourism outside the core film-related tourism attraction and site. The ways that the town implements the all-for-one tourism mode in its public and residential areas include the integration of film elements into the town's basic facilities and the representations of film elements in the public areas, building the tourist-accessible film-related facilities and activities, and the encouragement of residents' participation in the film-related tourism businesses. By doing these, the town is attempting to use 'all' available landscapes, 'all' workable time, 'all' relevant industries and 'all' potential people [1,2] to manage and develop its tourism industries Thus, the study of film-related tourism also offers a different perspective to research the model of all-for-one tourism.

Based on the case of Hengdian, this paper also evaluated the benefits and drawbacks of developing and managing film-related tourism in the all-for-one mode at a tourism destination. The benefits include the enrichment of tourism products and activities and tourists' travel routes, the participation of more local people in the tourism and tourism-related industries, and the feasibility and universality of all-for-one film-related tourism at different kinds of tourism destinations. The drawbacks include the commercialisation of the destination itself and the spatial and labour displacement at the destination. In this regard, this study's significance is twofold—it contributes to the existing body of theoretical research and empirical investigation of film-related tourism. Additionally, the outcomes of this research are valuable for both film-related tourism literature and tourism practitioners in terms of implementing the all-for-one tourism model in a practical setting. Proposed and introduced by Chinese tourism practitioners in 2016, all-for-one tourism is now formally established in China. Hengdian is one of the Chinese destinations that develops its film-related tourism by applying the all-for-one tourism mode. It is important to note that all-for-one tourism is still a very new cultural phenomenon that is worth further researching in depth in future. The case of Hengdian in this paper contributes to understanding the ways the model of all-for-one tourism can be applied by a tourism destination, the benefits

and drawbacks of this tourism model, and the reasons why the Chinese government has promoted the development of all-for-one tourism at the nationwide level since 2016.

This study's limitations arise from the fact that its findings and discussions predominantly rely on a specific case, and as such, may not be entirely generalizable to other cases. Further research is necessary to validate the findings and to examine their applicability in different contexts. In addition, the discussion in this paper was based on the data collected from the ethnographic methods. For future research, all-for-one film-related tourism can also be studied by conducting quantitative research methods. A deeper understanding of how film-related tourism can be developed and managed within the all-for-one tourism model could have significant implications. It could help to provide a more complete and structured evaluation of all-for-one tourism, ultimately enhancing our comprehension of this approach to tourism.

**Funding:** This research received no external funding.

**Institutional Review Board Statement:** The study was conducted in accordance with the Declaration of Helsinki, and approved by the School of the Arts Research Ethics Committee and the Central University Research Ethics Committees of the University of Liverpool (protocol code 4838 and date of approval: 21 October 2020).

**Informed Consent Statement:** Informed consent was obtained from all subjects involved in the study.

**Data Availability Statement:** The data presented in this study are available on request from the corresponding author. The data are not publicly available because this study signed a confidentiality pledge with the respondents.

**Acknowledgments:** The author would like to express her sincere gratitude to her supervisor Les Roberts from the Department of Communication and Media, University of Liverpool, for his constant and valuable contributions to the author's research and his kind encouragement, expert supervision and excellent guidance throughout the author's research process. The author also would like to thank the anonymous reviewers and editors for their helpful suggestions for the improvement of this paper.

**Conflicts of Interest:** The author declares no conflict of interest.

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
