# Peer review of "Developing and Managing Film-Related Tourism in the All-for-One Model at a Tourism Destination: The Case of Hengdian Town (China)"

_tourismhosp, doi:10.3390/tourhosp4040034_

Round 1
Reviewer 1 Report
Comments and Suggestions for Authors
Review of “Developing and managing film-related tourism in the all-for-one model at a tourism destination, the case of Hengdian Town (China).”
This is an important paper which should be published. There is very little literature (in English) on the Chinese “all-for-one tourism” plan, which is still under a decade old but has been applied to hundreds of destinations. All-for-one tourism is very important for the continued growth of China’s tourism industry, and may well be spreading to other areas of the world. As the authour states, theirs is limited to one – very important – destination, but maybe it will stimulate further research and comparative papers on other kinds of destinations and, eventually, a comparative general assessment. They did not seem to have made any attempt to bring in other cases (the reviewer has visited another major media studio, and has focused on all-for-one tourism in examining metropolitan tourism in other cities in China).
The paper presents its findings clearly and covers most of the relevant literature. They might have looked at He, Liang, Li, Liu and Liu (2019) to see an example of how the use of mass data has been adapted for access and control of the many kinds of enterprises which must be coordinated in an all-for-one tourism region. Although the authours claim to have carried out ethnographic research, it is of a rather slim nature – though well done as far as it goes. Ethnography usually involves living with or working with the people being studied, not just visiting and asking questions: for instance someone might have work or volunteered – in the film industry, in the hospitality industry or even as a tour guide, to get more in depth “insider” views, but what they did do is a good start, especially for the general industrial side of Hengdian. In the reviewer’s experience in a major media studio region in Shandong, s/he witnessed much more direct interactions of tourists with the media processes, especially kinds of Cosplay: children dressed up and acting out, adults wearing the clothing of actors or characters or other historical periods and different nations. Nothing like this was mentioned for Hengdian, which is another reason for my guess that this ethnography is a ‘bit thin’?!!!
The authours have quite strongly shown us the workings of this tourism system in Hengdian and its successful penetration of other tourist-related enterprises and even private investment decision making. They have also wisely considered the dis-advantages of this plan, focusing on over-commercialization such that some residents are squeezed out. I’m not sure whether those people just found rents too expensive or whether they may have had poorer paying or no employment? The authours might also have discussed seasonality, and the relative devaluation of parts of the town/region that didn’t cater to the film industry. There may also be downturns in the film industry (as in the USA) which would devastate almost everybody, whereas some other tourism is based on spectacular monuments or natural features which don’t “go away.”
He, Dian, Ying Liang, Xiaolong Li, Yao Liu & Jianglian Liu
2019 “Systematic Framework of the All-for-One Tourism Digital Ecosystem.” First Online: 13 September 2019, Part of the Communications in Computer and Information Science book series (CCIS volume 1059)
Author Response
I have provided a point-by-point response to the reviewer's comments, please see the attachment.

Reviewer 2 Report
Comments and Suggestions for Authors
The paper is one of a well-written but has several minor mistakes which are shown at the attached paper.

Author Response
I have provided a point-by-point response to the reviewer’s comments, please see the attachment.

Reviewer 3 Report
Comments and Suggestions for Authors
Dear Author,
Your paper exhibits relevance and adherence to scientific writing norms, although it necessitates minor revisions to elevate its overall quality. My observations and recommendations are outlined below:
In the title, I recommend to replace "the film-related tourism" by "film-induced tourism" .
The abstract is well-structured. The authors are invited to provide a brief description of the methodology used.
The format of the in-text citations you have used is not appropriate. Please refer to author guidelines.
The research method adopted is relevant and aligns with the paper's objectives, but it's not well-described. The authors are encouraged to justify the techniques and approaches used and explain their suitability within the research context.
Heading 4 should start by Results, in this case, the title of this heading should be "Results and Discussion" instead of "Discussion and results"
The results are well-presented. The discussion section should include a discussion related to the obtained results with necessary comparisons.
The conclusion should encompass the main contributions of your research in advancing knowledge in the field.
There are too few references for such a hot topic. The authors should add more pertinent and recent papers to enhance the overall quality.
Author Response

(The authors gave the same response as above.)

Round 2
Reviewer 3 Report
Comments and Suggestions for Authors
The paper's quality was improved, it can be published in its current form.